# Chlorobenzene Removal Using DBD Coupled with CuO/γ-Al₂O₃ Catalyst

**Xingpeng Jin [1]** , **Guicheng Wang [2]**, **Liping Lian [1]**, **Fan Gao [1]**, **Renxi Zhang [1],***, **Weixuan Zhao [1,3],***, **Jianyuan Hou [1]**, **Shanping Chen [4]** and **Ruina Zhang [4]**

[1] Shanghai Key Laboratory of Atmospheric Particle Pollution and Prevention (LAP3), Institute of Environmental Science, Fudan University, Shanghai 200433, China; 19210740029@fudan.edu.cn (X.J.); lipinglian@fudan.edu.cn (L.L.); 19210740068@fudan.edu.cn (F.G.); 13916733297@126.com (J.H.)

[2] Institute of Developmental Biology and Molecular Medicine, School of Life Sciences, Fudan University, 220 Handan Rd., Shanghai 200433, China; wangguicheng@fudan.edu.cn

[3] Fujian Key Laboratory of Architectural Coating, Skshu Paint Co., Ltd., 518 North Liyuan Avenue, Licheng District, Putian 351100, China

[4] Shanghai Institute for Design & Research on Environmental Engineering, Shanghai 200232, China; shanpch@163.com (S.C.); zhangrn@huanke.com.cn (R.Z.)

* Correspondence: zrx@fudan.edu.cn (R.Z.); 15210740032@fudan.edu.cn (W.Z.)

**Abstract:** The removal of chlorobenzene using a dielectric barrier discharge (DBD) reactor coupled with CuO/γ-Al₂O₃ catalysts was investigated in this paper. The coupling of CuO enhanced the chlorobenzene degradation and complete oxidation ability of the DBD reactor, especially under low voltage conditions. The characterization of catalyst was carried out to understand the interaction between catalyst and plasma discharge. The effects of flow rate and discharge power on the degradation of chlorobenzene and the interaction between these parameters were analyzed using the response surface model (RSM). The analysis of variance was applied to evaluate the significance of the independent variables and their interactions. The results show that the interactions between flow rate and discharge power are not negligible for the degradation of chlorobenzene. Moreover, based on the analysis of byproducts, 4-chlorophenol was discriminated as the important intermediate of chlorobenzene degradation, and the speculative decomposition mechanism of chlorobenzene is explored.

**Keywords:** plasma catalysis; dielectric barrier discharge (DBD); chlorobenzene; chlorinated volatile organic compounds (CVOCs)

## 1. Introduction

Chlorinated volatile organic compounds (CVOCs) exist in various environmental matrices such as air, water, sand, clay, and sludge [1–3]. Due to its high volatility and strong recalcitrance to degradation, CVOCs are more toxic than other VOCs and their environmental lifetime is longer [4,5]. Therefore, most CVOCs have been listed as priority pollutants by many countries with strict monitoring. However, CVOCs have been widely used in industry as organic solvents for processes such as metal degreasing and dry cleaning [6–8]. Various commercial products containing CVOCs and the main sources of CVOCs principally include industrial emissions, the consumption of CVOC-containing products, the disinfection process, as well as improper storage and disposal operation [9]. Thus, the control of CVOCs has drawn the attention of many investigations.

In recent decades, the study of CVOCs control mainly focused on the adsorption method [10–15], photo-degradation [16–18], combustion treatment [19,20], biodegradation [21,22] and catalytic oxidation [23–27]. However, the adsorption method only presents a temporary pollution transfer from the gaseous to the solid phase, requiring further treatment for the adsorbents. The use of photo-degradation is limited by the stability of the light

source and slow reaction rate, while the combustion treatment requires high combustion temperatures as well as huge energy consumption. The catalytic oxidation method is also limited by a high and narrow temperature window for the complete treatment of CVOCs. Moreover, some studies indicated that the catalytic oxidation of chlorobenzene in high temperatures might produce undesired products such as polychlorinated benzene [28,29]. As a destructive method, the efficiency of non-thermal plasma technology (NTP) treating VOCs has been proved by many studies [30–32]. However, the application of NTP is limited by its drawbacks, such as low selectivity and the production of unwanted products. Therefore, many studies focus on the hybrid technologies of NTP coupled with other technologies [33–36]. Among these hybrid systems, plasma catalysis technology shows its advantages among the treatments of typically VOCs for its high selectivity and removal rate because the gas-phase reaction and catalyst surface reaction induces a synergistic effect of plasma and catalysts [37–41]. In previous studies, the enhancing effect of various transition metal oxides for the treatment of VOCs by NTP has been confirmed.

Among them, copper-based catalysts have received extensive attention due to their relatively low cost and high reactive activity. Zhu et al. reported that the supported copper oxide catalyst showed the optimal acetone degradation ability in a catalyst-enhanced DBD system [42]. In another study, the supported copper oxide catalyst exhibits a better performance assisted NTP system for benzene treatment among five transition metal oxides [43]. Moreover, the copper-based catalysts are also capable of catalyzing the CO oxidation reaction, which helps to improve the $CO_2$ selectivity of VOC degradation. Feng et al. used catalytic CuO nanowires enhanced by plasma catalyzing CO oxidation [44]. Those results showed that on the surface of copper-based catalysts, $Cu^{2+}$ ions not only act as a catalyst for the oxidation of CO but also act as a reducing agent for CO oxidation under oxygen-poor conditions. During the plasma treatment, some of the $Cu^{2+}$ ions can be partially reduced to $Cu^+$ ions, which has a stronger catalytic ability when catalyzing CO oxidation.

Although the capacity of copper-based catalyst-assisted cooperative plasma treatment of VOC has been proved [45–47], there are few reports focus on copper-based catalyst-assisted plasma treatment of CVOC [48]. In addition, previous studies have found that either single plasma treatment [34] or catalyst degradation of chlorobenzene [28,29] may produce some byproducts that are more toxic than the original pollutants. Therefore, it is of great significance to identify the reaction products of hybrid technology, and it is also necessary to understand the interaction of different process parameters and their influence on the reaction products.

To better understand the interaction between different input variables, in recent years, response surface methodology (RSM) has attracted the attention of process research and optimization. RSM is a statistical model based on the design of experiments [49]. It considers the nonlinear relationship between multiple inputs and multiple output variables. Through experimental design and model building, it helps to predict the importance and interaction of independent variables and optimize the performance of complex systems. So far, limited efforts were made to investigate plasma processes by the RSM model [50,51].

Due to the toxicity and handling difficulty of CVOCs, in this paper, chlorobenzene, the most basic chlorinated aromatic hydrocarbon, was selected as the substrate. The improved effects of the DBD reactor coupled with $CuO/\gamma$-$Al_2O_3$ catalysts for chlorobenzene degradation were studied. The removal efficiency and CO and $CO_2$ selectivity of chlorobenzene and the production of $O_3$ after discharge were investigated. The response surface model is used to analyze the relationship between the main process parameters. Furthermore, the byproducts after plasma-catalytic treatment were analyzed to speculate the possible reaction mechanism.

## 2. Experimental Section

### 2.1. Preparation of Catalysts

The $CuO/\gamma$-$Al_2O_3$ catalyst was prepared by the excessive impregnation method. The amount of precursor ($Cu(NO_3)_2 \cdot 3H_2O$) necessary to obtain the copper nitrate solution with a concentration of 1 mol/L was dissolved in distilled water, and $\gamma$-$Al_2O_3$ powder was added to the excess volume of copper nitrate solution, followed by ultrasonic oscillation for 20 min, immersion for 24 h and filtration. The slurry obtained by filtration was dried at 60 °C for 12 h. The samples were then calcined at 500 °C for 5 h. For the preparation of pure $\gamma$-$Al_2O_3$ powder, distilled water was used instead of copper nitrate solution. The $CuO/\gamma$-$Al_2O_3$ catalyst and pure $\gamma$-$Al_2O_3$ powder were ground and sieved to 40–60 meshes for further experiments.

The Cu loading of catalyst was measured by ICP-OES (Thermo iCAP 7400, Thermo Fisher Scientific, Waltham, MA, USA). The Brunauer–Emmett–Teller (BET) surface area of catalyst was measured by $N_2$ adsorption and desorption experiments using Micromeritics Tristar II 3020 instrument (Micromeritics, Norcross, GA, USA). The phase structure of the catalyst was characterized by an X-ray diffractometer (XRD, Bruker D8 Advance, Bruker, Billerica, MA, USA). The Cu K$\alpha$ radiation ($\lambda$ = 0.1542 nm) is operated at 40 kV and 40 mA with a 2$\theta$ range of 10°–80° to obtain the XRD patterns. The oxidative states of Cu before and after the discharge were detected by an X-ray photoelectron spectroscopic (XPS) spectrometer (Thermo Scientific K-Alpha). The pass energy of the full spectrum scan is 100 eV with a step length of 1 eV. The pass energy of the narrow-spectrum scan is 50 eV, with a step length of 0.05 eV.

### 2.2. Experimental Setup

The experimental setup consisted of three parts: mixed gas generator, DBD reactor, and outlet gas detector (Figure 1). Chlorobenzene gas flow was obtained by feeding dry $N_2$ gas to a chlorobenzene bubbling bottle. The temperature of the bottle was maintained at 25 °C by a water bath. The chlorobenzene flow and $O_2$ gas flow were mixed in a mixing chamber before they were provided into the DBD reactor. The mass flow controllers were used to obtain the gas contained 200 ppm chlorobenzene with different gas flow rates ($1$–$5$ L·min$^{-1}$).

The DBD reactor (Figure 2) used in this experiment consisted of two coaxial cylinders. The inner quartz glass cylinder, filled with graphite powder with an outer diameter of 20 mm and a thickness of 1.5 mm, served as the inner electrode and ground electrode of the reactor. The outer quartz glass tube, surrounded by the copper foil with an inner diameter of 26 mm and a thickness of 1.5 mm, served as the outer electrode and the high-voltage electrode of the reactor. The copper wire with a diameter of 2 mm was used to connect the high-voltage alternating current (AC) power supply and attach the copper foil tightly to the quartz glass tube. The length of copper foil is 40 mm, creating a coaxial cylinder discharge zone 0.5 g $CuO/\gamma$-$Al_2O_3$ catalyst (or $\gamma$-$Al_2O_3$ powder) was packed in the catalyst zone with an outer diameter of 26 mm, an inner diameter of 20 mm, and a height of 2 mm.

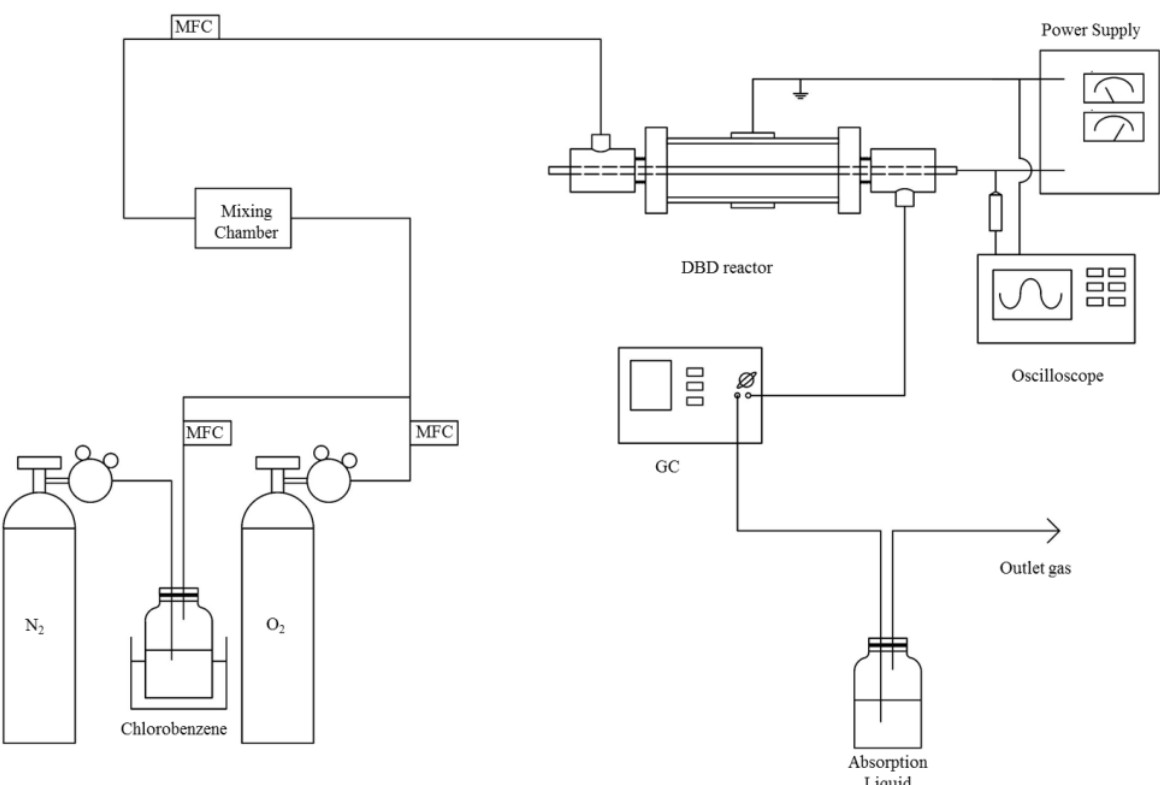

**Figure 1.** Schematic diagram of the experimental setup (outlet).

The reactor was powered by an AC power with a frequency of 10–20 kHz and a voltage of 3.6–6 kV. The discharge parameters and Lissajous figure of the experiment were obtained by a 200 MHz digital phosphor oscilloscope (Tektronix, TDS2024B, Beaverton, OR, USA) using a 1000:1 high-voltage probe (Tektronix P6015A, Beaverton, OR, USA) and 0.47 μF equivalent capacitor.

The specific energy density (SED; $J \cdot L^{-1}$) of plasma is defined as the energy obtained per volume of gas:

$$\text{SED} = \frac{P}{Q} \times 60 \tag{1}$$

where P (W) is the discharge power of the DBD reactor, and Q ($L \min^{-1}$) is the gas flow rate.

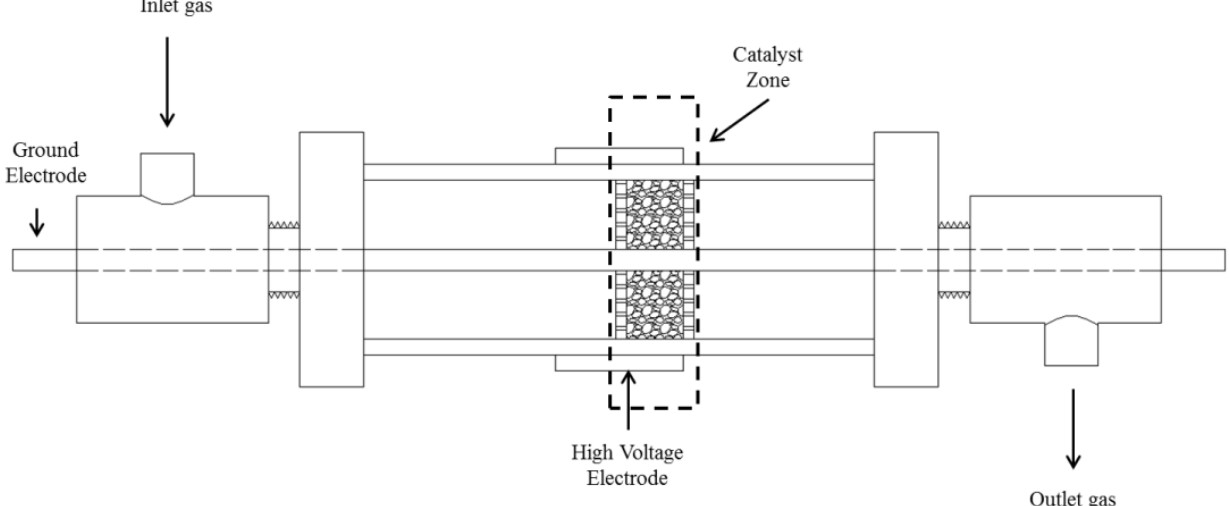

**Figure 2.** Schematic diagram of the DBD reactor.

In this experiment, the composition of mixed gas before and after discharge was analyzed online by a gas chromatograph (GC9790, FULI Instruments, Wenling City, Zhejiang, China) for chlorobenzene and a gas chromatograph (GC930, Haixin Instruments, Shanghai, China) for CO and $CO_2$. The removal efficiency, CO, $CO_2$, and $CO_x$ selectivity of chlorobenzene were defined as:

$$\eta(\%) = \frac{[CB]_{in} - [CB]_{out}}{[CB]_{in}} \times 100\% \tag{2}$$

$$CO\ selectivity(\%) = \frac{[CO]}{([CB]_{in} - [CB]_{out}) \times 6} \times 100\% \tag{3}$$

$$CO_2\ selectivity(\%) = \frac{[CO_2]}{([CB]_{in} - [CB]_{out}) \times 6} \times 100\% \tag{4}$$

$$CO_x\ selectivity(\%) = \frac{[CO_2] + [CO]}{([CB]_{in} - [CB]_{out}) \times 6} \times 100\% \tag{5}$$

where $[CB]_{in}$ and $[CB]_{out}$ are the inlet and outlet concentrations of chlorobenzene (ppm), and $[CO]$ and $[CO_2]$ are the outlet volumetric concentrations of CO and $CO_2$ (ppm).

The $O_3$ concentration of outlet gas was determined by the iodometric method. The collected gas was pumped into a gas scrubber filled with potassium iodide solution with a speed of 500 mL/min for 10 min. Then 1~2 mL of dilute sulfuric acid was used to adjust the pH of absorption solution below 2. After standing for 5 min, a few drops of the starch solution was added into the absorption solution as an indicator, and the sodium thiosulfate standard solution with a concentration of 0.02 mol/L was used to titrate the absorption solution. The byproducts of chlorobenzene degradation after single plasma treatment and plasma catalysis treatment with an applied voltage of 4.2 kV, flow rates of 3 $L \cdot min^{-1}$ and SED of 950 $J \cdot L^{-1}$ were analyzed by a gas chromatography-mass spectrometer (GC-MS).

To exclude the extra effect of catalyst (adsorption effect, etc.) on chlorobenzene removal, the experiment was carried out after the catalyst bed reached adsorption equilibrium (the concentration of chlorobenzene remains steady after passing through the reactor). The effect of temperature is negligible since the control experiment carried out at 105 °C (the highest temperature detected during the experiment) shows that there is no significant degradation of chlorobenzene without discharge.

In this work, discharge power and gas flow rate, the most important process parameters for plasma discharge, were selected as input parameters, while chlorobenzene removal efficiency and $CO_2$ selectivity were selected as response parameters. The input parameters were standardized by z-score. To ensure the comparability between the action coefficients:

$$x_i = \frac{X_i - \overline{X}}{\sigma} \tag{6}$$

where $x_i$ is the standardized value of the ith variable, $X_i$ is the real value of the ith variable, $\overline{X}$ is the average value of data, and $\sigma$ is the standard deviation of the data. The standardized and real values of the selected plasma processing parameters were given in Table 1.

**Table 1.** Schematic diagram of the DBD reactor. The chosen process parameters and their standardized values.

| Parameters | | | | | |
|---|---|---|---|---|---|
| Flow Rate (L/min) | 1 | 2 | 3 | 4 | 5 |
| $x_1$ | −1.46 | −0.73 | 0 | 0.73 | 1.46 |
| Discharge Power (W) | 37.5 | 44.3 | 55 | 60 | 67.5 |
| $x_2$ | −1.28 | −0.73 | 0.15 | 0.56 | 1.17 |

In this study, a second-order polynomial response equation was established to correlate the relationship between independent plasma operation variables and responses variables:

$$Y = \beta_0 + \beta_1 x_1 + \beta_2 x_2 + \beta_{11} x_1^2 + \beta_{22} x_2^2 + \beta_{12} x_1 x_2 + \varepsilon \tag{7}$$

where Y is the response, $x_1$, $x_2$ are the standardized variables, $\varepsilon$ is the error, $\beta_0$ is a constant, $\beta_1$, $\beta_2$ are linear coefficients of two variables, $\beta_{11}$, $\beta_{22}$ are quadratic coefficients of two variables, and $\beta_{12}$ are interaction coefficients of two variables, respectively. To estimate the quality of fit and the significance of chosen parameters, the analysis of variance was performed, including the Fisher's F-test and the calculation of correlation coefficient ($R^2$) for the generated models [52].

### 3. Results and Discussion

*3.1. Preparation of Catalysts*

According to the results of ICP-OES, the copper content of $CuO/\gamma$-$Al_2O_3$ catalyst was 4.5 wt%. Table 2 shows the structural properties of the $CuO/\gamma$-$Al_2O_3$ catalyst and $\gamma$-$Al_2O_3$ powder. Because the introduction of Cu will cover part of the pores, the specific surface area and the pore volume of $\gamma$-$Al_2O_3$ powder decrease from 168.6 $m^2 \cdot g^{-1}$ to 141.3 $m^2 \cdot g^{-1}$ and from 0.95 $cm^3 \cdot g^{-1}$ to 0.72 $cm^3 \cdot g^{-1}$, respectively.

**Table 2.** Physicochemical properties of the $CuO/\gamma$-$Al_2O_3$ catalyst and $\gamma$-$Al_2O_3$ powder.

| Sample | BET Surface Area ($m^2/g$) | Total Pore Volume ($cm^3/g$) | Average Pore Diameter (nm) |
|---|---|---|---|
| $\gamma$-$Al_2O_3$ | 168.6 | 0.95 | 22.7 |
| $CuO/\gamma$-$Al_2O_3$ | 141.3 | 0.72 | 20.5 |

Figure 3 shows the XRD patterns of the $CuO/\gamma$-$Al_2O_3$ catalyst and the $\gamma$-$Al_2O_3$ powder. The formation of CuO crystals was distinguished according to the extra diffraction signals of the $CuO/\gamma$-$Al_2O_3$ catalyst at $2\theta = 35.6°$, $38.7°$, $48.7°$, $61.5°$, and $74.9°$ (JCPDS 48-1548) compared with the $\gamma$-$Al_2O_3$ powder.

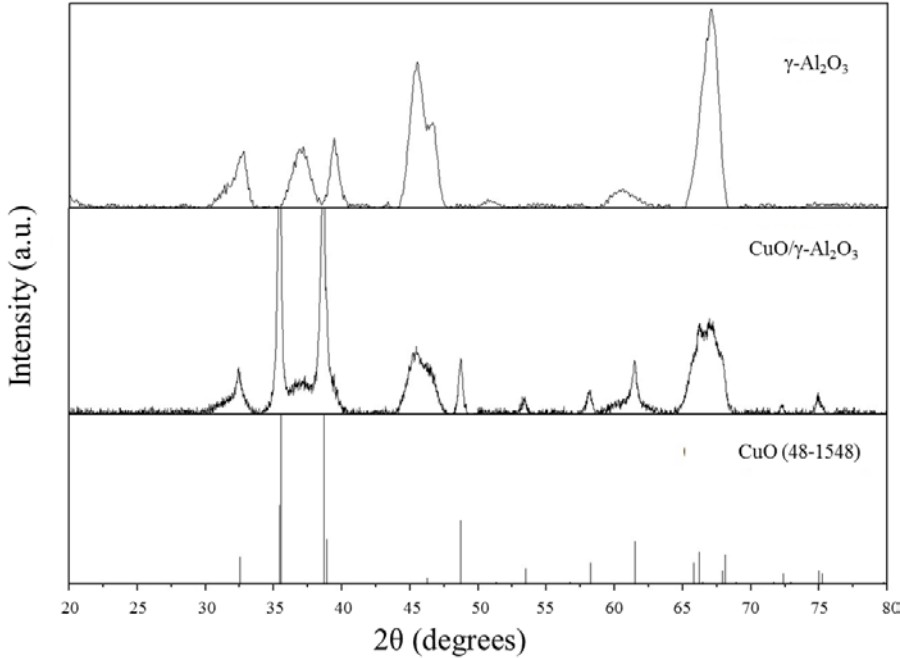

**Figure 3.** XRD patterns of the $CuO/\gamma$-$Al_2O_3$ catalysts and the $\gamma$-$Al_2O_3$ powder.

Figure 4 shows the XPS patterns of the Cu 2p in the CuO/γ-Al$_2$O$_3$ catalysts before and after the DBD discharge. The spin-orbit split on the 2p spectrum of Cu was observed. The 2p orbit of Cu before discharge was shown in figure (a). The binding energy of Cu (932.4 eV for Cu$_2$O and 933.9 eV for CuO) and the existence of the satellite peak suggest that the main chemical state of surface Cu is Cu$^+$(Cu$_2$O) and Cu$^{2+}$(CuO). The mass ratios of Cu$^+$(Cu$_2$O) and Cu$^{2+}$(CuO) are 23.1% and 76.9%, respectively. The results of XPS analysis indicate that the mixture of Cu$^+$ and Cu$^{2+}$ forms the upper layer of the CuO/γ-Al$_2$O$_3$ catalyst, and Cu$^{2+}$ is the main state of CuO/γ-Al$_2$O$_3$. The 2p orbit of Cu after DBD discharge was shown in Figure 4b. The binding energy of Cu (932.4 eV for Cu$_2$O and 934.6 eV for Cu(OH)$_2$) and the existence of the satellite peak suggest that the main chemical state of surface Cu is Cu$^+$(Cu$_2$O) and Cu$^{2+}$(Cu(OH)$_2$). The mass ratios of Cu$^+$(Cu$_2$O) and Cu$^{2+}$(CuO) are 81.6% and 18.4%, respectively. The increase in the Cu$^+$ ratio is mainly due to the reduction in Cu$^{2+}$ as an oxidant during the discharge process [44,53]. The production of Cu$^{2+}$ (Cu (OH)$_2$) is mainly due to the adsorption of water from the degradation of chlorobenzene on the catalyst surface.

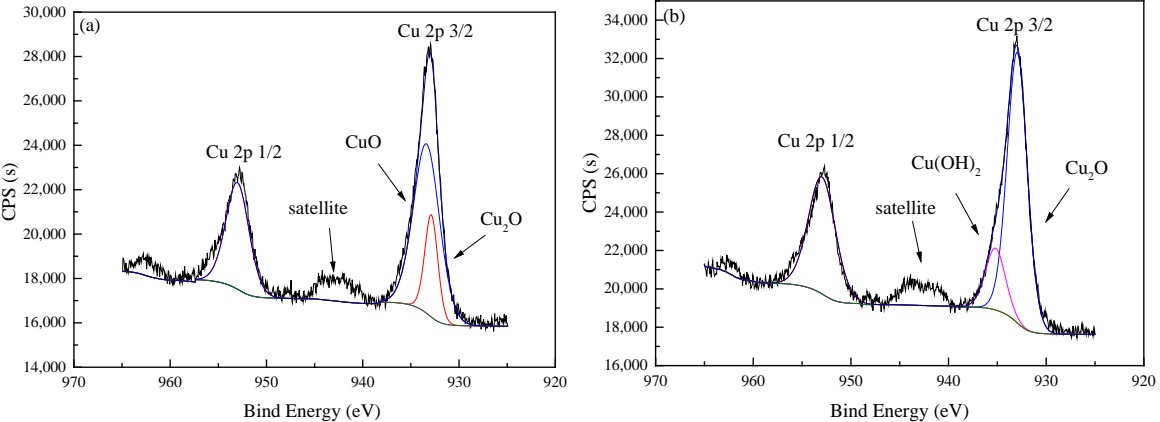

**Figure 4.** XPS spectra of the CuO/γ-Al$_2$O$_3$ catalyst before and after discharge: (**a**) before; (**b**) after.

### 3.2. Effect of SED on the Performance of DBD Reactor

3.2.1. Chlorobenzene Removal Efficiency

Figure 5 shows the removal efficiency of chlorobenzene under different SED in a single DBD reactor and two combined plasma-catalytic reactors. Compared with the other two types of plasma-catalytic reactors, the CuO packing reactor performed better chlorobenzene degradation (58.9%) when the SED is 750 J·L$^{-1}$, about 10% higher than that in the DBD reactor loaded with γ-Al$_2$O$_3$ powder (53.8%) and 66% higher than that in the single DBD reactor (32.9%). In the meanwhile, the increase in SED is beneficial for the chlorobenzenes to degrade. However, when the SED of plasma exceeds 1000 J·L$^{-1}$, this enhancing effect of CuO loading is weakened. As SED reached 1350 J·L$^{-1}$, over 90% of chlorobenzene removal can be achieved by these reactors, and the differences among these reactors are ignorable.

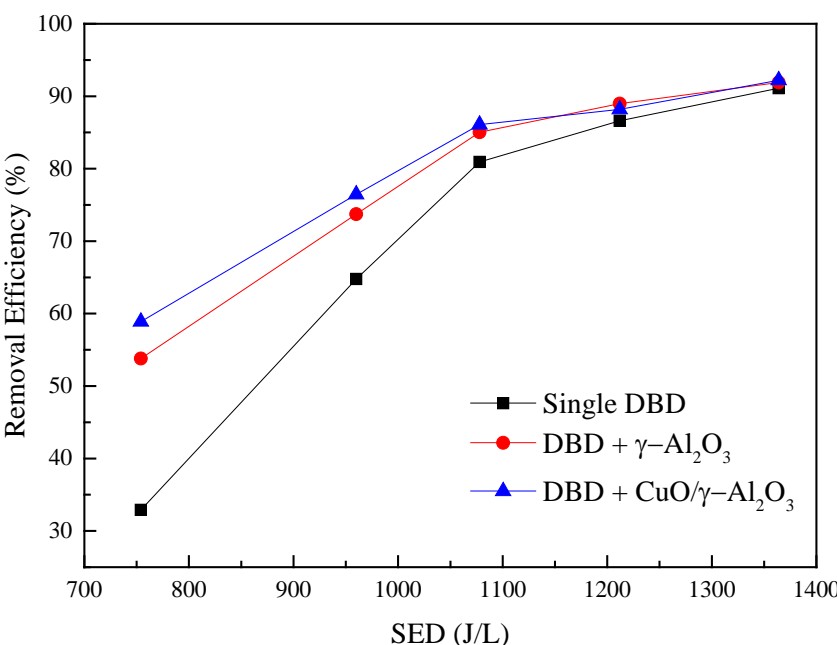

**Figure 5.** Removal efficiency of chlorobenzene with catalyst loadings and absence of catalyst as a function of SED (flow rate = 3 L/min).

### 3.2.2. $CO_x$ Selectivity

Figure 6 shows the $CO_x$ selectivity of chlorobenzene degradation versus SED in a single DBD reactor and two combined plasma-catalytic reactors. The CuO packing reactor obtained the highest $CO_x$ selectivity of chlorobenzene among the three tested reactors. Moreover, the total $CO_x$ selectivity of chlorobenzene in the CuO packing reactor changes slowly with the increase in SED, while the total $CO_x$ selectivity of chlorobenzene in the other two reactors turns to decrease with the increase in SED. It can be concluded that CuO loading has an improved effect on the oxidation of chlorobenzene. However, further analysis shows that the steady $CO_x$ selectivity of chlorobenzene in the CuO packing reactor is the result of the decreasing $CO_2$ selectivity and increasing CO selectivity with the increase in SED, which indicates that $CuO/\gamma$-$Al_2O_3$ catalysts might promote the degradation of chlorobenzene by oxidizing CO to $CO_2$. Satoshi studied the reaction mechanism of CO oxidation on Fe-Cu/$CeO_2$catalysts and suggested that the $Cu^+$ species serve as CO chemisorption site, and the reaction was preceded via the Mars–van Krevelen mechanism [54]. Connecting with the results of Sections 3.1 and 3.2.1, the $CuO/\gamma$-$Al_2O_3$ catalyst might promote the degradation of chlorobenzene by oxidizing CO to $CO_2$ through the Mars–van Krevelen mechanism. CO is oxidized by the $Cu^{2+}$ on the surface of the catalyst while $Cu^{2+}$ is reduced to $Cu^+$, then $Cu^+$ will be oxidized by other active particles such as OH· and O·. However, the reactive site on the $CuO/\gamma$-$Al_2O_3$ catalyst is limited; as SED increases, more CO needs to be oxidized by $Cu^{2+}$ to maintain the $CO_2$ selectivity of chlorobenzene degradation. For this study, since most of $Cu^{2+}$ is reduced to $Cu^+$ after the discharge (Section 3.1), the decline of $CO_2$ selectivity starts with the increase of SED. Furthermore, since the enhancing effects of CuO loading for the oxidation of CO are weakened as SED increases, the enhancing effects of CuO loading for the degradation of chlorobenzene will show the same trend (Section 3.2.1).

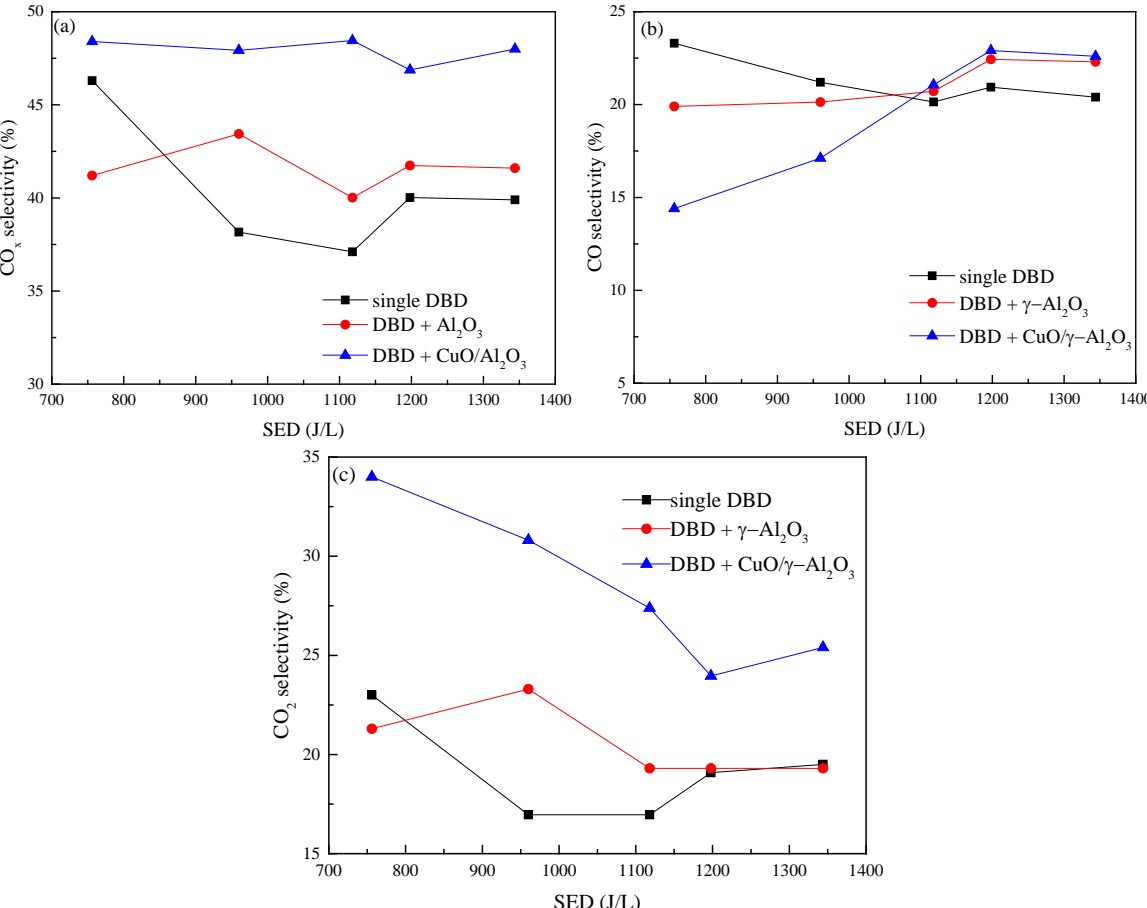

**Figure 6.** $CO_x$ (**a**), CO (**b**), and $CO_2$ (**c**) selectivity of chlorobenzene with catalyst loadings and absence of catalyst as a function of SED (flow rate = 3 L/min).

### 3.2.3. Ozone Production

The production of ozone is one of the most significant evaluation indexes for the DBD discharge. Figure 7 shows the variation of $O_3$ production with different SED in a single DBD reactor and the plasma-catalytic reactors. It can be seen that the addition of $CuO/\gamma$-$Al_2O_3$ in the DBD reactor reduces the production of ozone when the SED is less than $750 \ J \cdot L^{-1}$, but as SED increases to $900 \ J \cdot L^{-1}$ or higher, the addition of $CuO/\gamma$-$Al_2O_3$ in the DBD reactor turns to improve the production of ozone. Chen also found that catalysts behave conversely when placed after or inside the reactor [55]. The contrary effect of $CuO/\gamma$-$Al_2O_3$ loading under different SED indicates that the loading of $CuO/\gamma$-$Al_2O_3$ might influence the production of ozone by different mechanisms. On the one hand, the oxidation of chlorobenzene induced by CuO catalyst expands the consumption path of active oxygen. It was considered that the addition of a catalyst could provide adsorption sites for active oxygen [43]. The absorbed oxygen can be consumed through the oxidation of $Cu^+$ (Section 3.2) or the reaction of chlorobenzene (Section 3.5). Moreover, considering the oxidation potential of ozone (2.07 eV), ozone can also be consumed through the oxidation process of chlorobenzene [56]. On the other hand, it was found that the coupling of the catalyst can improve the discharge condition of the reactor, which is beneficial for the production of active oxygen [57]. As is known, the production of active oxygen is related to the formation of ozone. The contrary effect of $CuO/\gamma$-$Al_2O_3$ loading for the production of active oxygen ultimately makes the behavior of $CuO/\gamma$-$Al_2O_3$ loading changes under different SED.

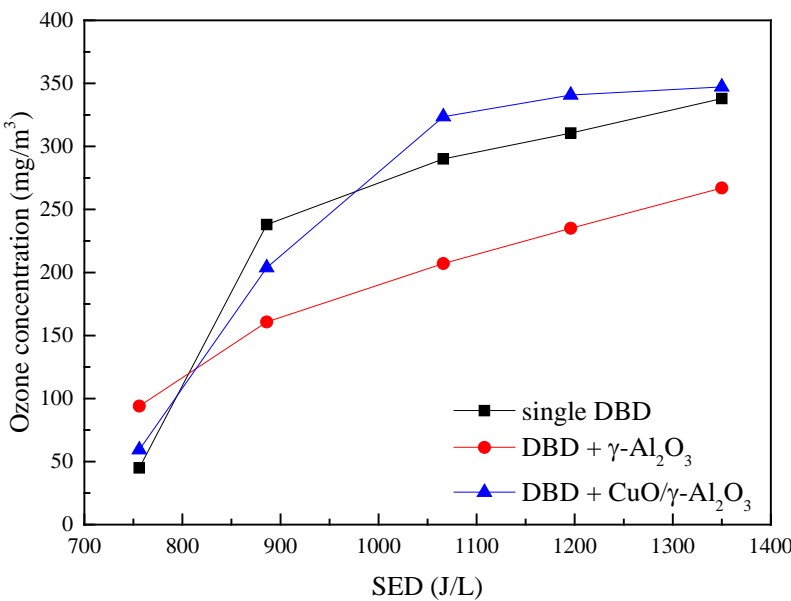

**Figure 7.** Effect of SED on the concentration of $O_3$ in the single DBD reactor and the combined plasma catalysis reactor.

### 3.3. Effect of Flow Rate on the Performance of DBD Reactor

Figure 8 shows the degradation efficiency of chlorobenzene at different flow rates while the SED of plasma is set to be 800 J·L$^{-1}$. For the DBD reactor coupled with CuO/$\gamma$-Al$_2$O$_3$ catalyst, within the tested flow rate (3 to 5 L·min$^{-1}$), the removal efficiency of chlorobenzene increased gradually with flow rate. For the single DBD reactor and the DBD reactor packed with $\gamma$-Al$_2$O$_3$ powder, the removal efficiency of chlorobenzene increased significantly with flow rate when the flow rate is low (3 to 4 L·min$^{-1}$). However, under a high flow rate (4 to 5 L·min$^{-1}$), the removal efficiency of chlorobenzene in these reactors appeared to fluctuate slowly with flow rate.

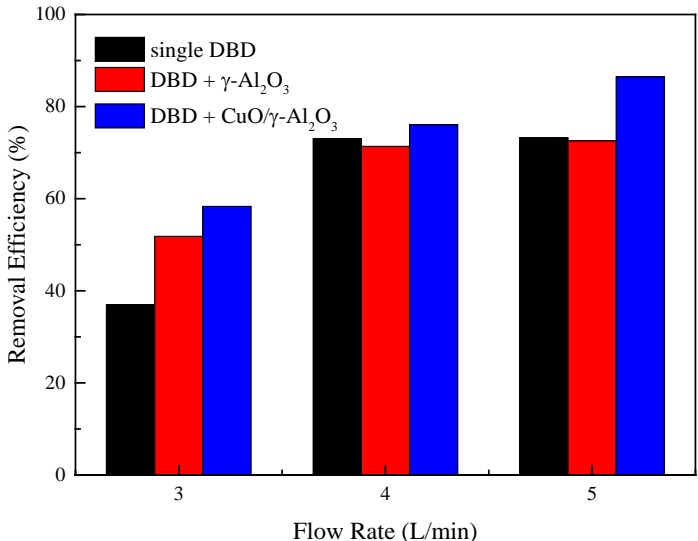

**Figure 8.** Effect of flow rate on the removal efficiency of chlorobenzene SED = 800 J·L$^{-1}$.

In the previous studies, Li et al. found that the removal efficiency of VOCs will gradually decrease with the increase in flow rate with fixed discharge power, which is attributed to the longer residence time at a low flow rate [58]. In this study, it was found that the increase in flow rate is beneficial for the removal of the chlorobenzene when

the discharge was carried with the same SED. Thus, for per volume of gas, the total energy of electronics will not change with flow rate. Instead, the amount and energy of electronics will be influenced by different flow rates: with the increased flow rate and discharge power, the amount of electronics will decrease, and the energy of electronics will increase. As is known, the function of DBD discharge requires the break of chemical bonds under the attack of high-energy electronics. If the energy of electronics (proportional to discharge power) is not high enough, the possibility for the break of chemical bonds will be small, and the energy will be wasted. For the discharge carried out with a higher flow rate and voltage, the energy efficiency can be improved since the waste of energy was avoided. However, when the break of chemical bonds (production of active radicals) is not the reaction controlling step, the enhancing effect of energy efficiency improvement will be weakened (single DBD reactor). The beneficial effects of catalyst loading can be summarized into two parts: 1. reducing the energy required for the break of chemical bonds by the adsorption sites, which helps to improve the energy efficiency of the reactor (better performance of $\gamma$-$Al_2O_3$ and $CuO/\gamma$-$Al_2O_3$ loading when the flow rate is 3 L·min$^{-1}$); 2. providing more consumption paths for active radicals, which helps to use the improve of energy efficiency (better performance of $CuO/\gamma$-$Al_2O_3$ loading when the flow rate is 5 L·min$^{-1}$).

### 3.4. RSM Analysis

The regression model was used in this study to analyze the removal efficiency and $CO_2$ selectivity of chlorobenzene in the DBD reactor packed with $CuO/\gamma$-$Al_2O_3$ catalyst. Figures 9 and 10 show the chosen process parameters and corresponding results of chlorobenzene degradation. The removal efficiency of chlorobenzene by plasma catalysis varied between 51.5% and 97.4%, and $CO_2$ selectivity varied between 21.3% and 33.3%. The fitted models of removal efficiency and $CO_2$ selectivity in terms of standardized input parameters are given as:

Removal efficiency:

$$Y_1 = 80.8 - 7.35x_1 + 11.49x_2 - 0.0668 {}^* x_1{}^2 + 2.66x_1x_2 - 1.35x_2{}^2$$

$CO_2$ selectivity:

$$Y_2 = 26.1 - 2.18x_1 - 2.62x_2 + 0.337{}^* x_1{}^2 - 0.410x_1x_2 + 0.562x_2{}^2$$

Table 3 shows the analysis of variance for the fitted models. The results show that the response of the model is significant because the F values of $Y_1$ and $Y_2$ are 62.23 and 82.25; for comparison, the critical value of this model is 6.61. Moreover, the ultimate low *p*-value (<0.01) supports the hypothesis that the goodness-of-fit is high (confidence level >95%). It is generally certified that most of the variation in the response can be explained by the fitted model with F-value higher than the critical value and a *p*-value lower than 0.01. The root means square errors of $Y_1$ and $Y_2$ were 2.632 and 1.016, respectively, which were comparably smaller than the predicted values. The regression correction ($R^2$) coefficients ($Y_1$ = 0.9766, $Y_2$ = 0.9466, close to 1) also indicates that the regression model fits the experimental results well. The coefficient of variation (C.V.), the ratio of the standard error of the estimated value to the average value of the response, can serve as the index judging the reproducibility of the model. For this experiment, the C.V. value of $Y_1$ (<10%) and $Y_2$ (<10%) suggests that the reliability and reproducibility of the model are acceptable [51].

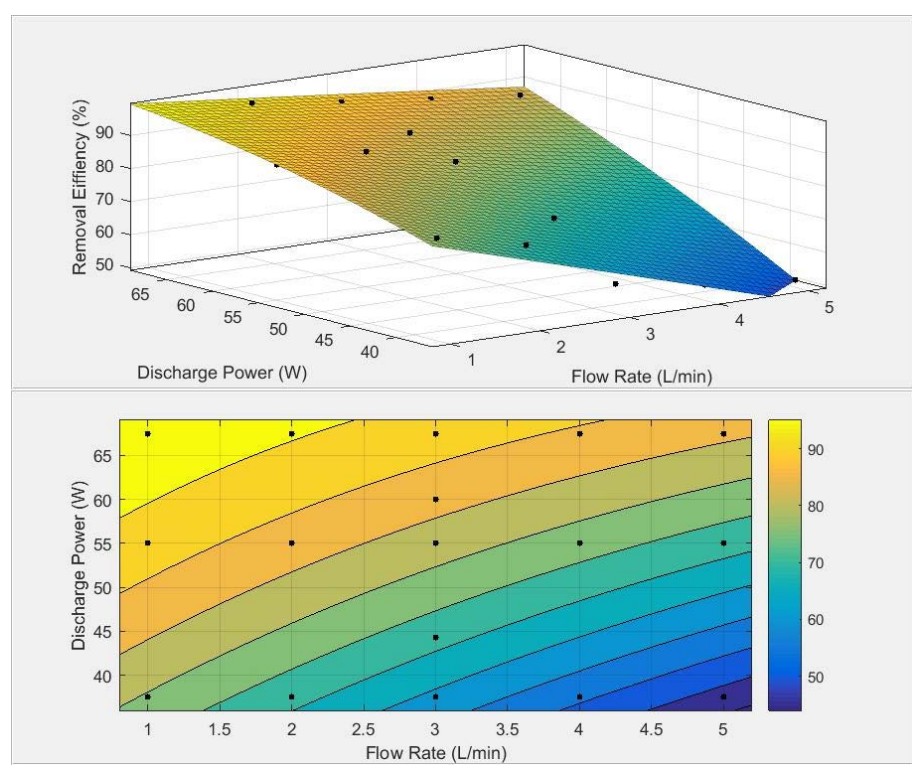

**Figure 9.** Effect of flow rate and discharge power on efficiency of chlorobenzene in the Cu/γ-Al$_2$O$_3$ reactor.

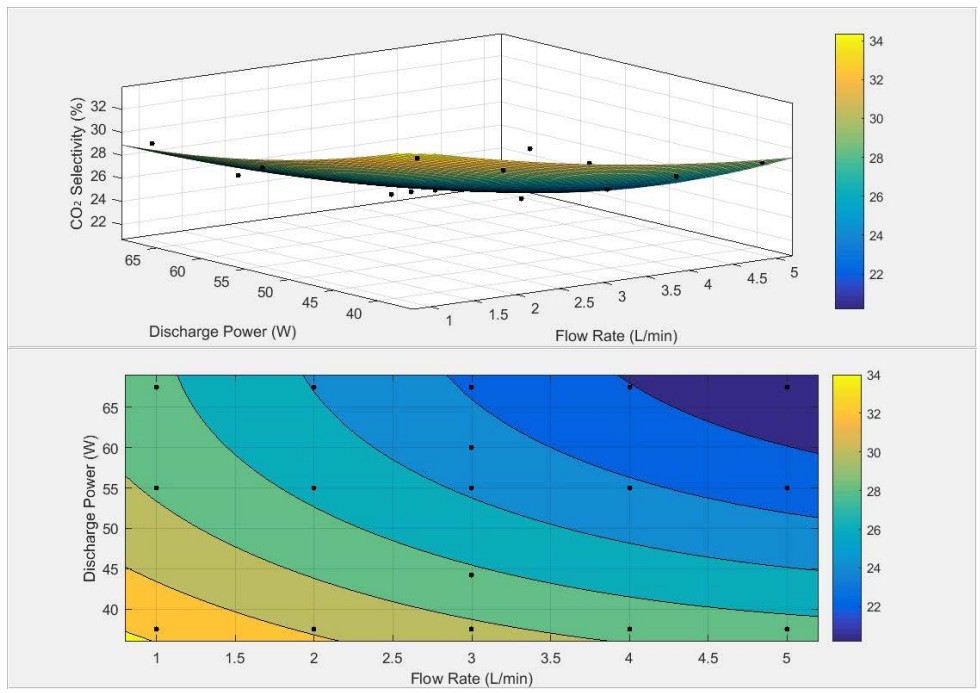

**Figure 10.** Effect of flow rate and discharge power on CO$_2$ selectivity of chlorobenzene in the Cu/γ-Al$_2$O$_3$ reactor.

**Table 3.** The magnitude and significance analysis of factor effects on the responses.

| Response | Model Terms | Sum of Square | Degree of Freedom | Mean Square | F-Value | *p*-Value |
|---|---|---|---|---|---|---|
| Removal efficiency | Model | 3184.30 | 5 | 636.86 | 91.27 | <0.0001 |
| | $x_1$ | 864.36 | 1 | 864.36 | 123.88 | <0.0001 |
| | $x_2$ | 2112.32 | 1 | 2112.32 | 302.74 | <0.0001 |
| | $x_1^2$ | 0.062 | 1 | 0.062 | 0.01 | 0.9266 |
| | $x_1 x_2$ | 114.44 | 1 | 114.44 | 16.40 | 0.0019 |
| | $x_2^2$ | 14.90 | 1 | 14.9 | 2.14 | 0.1719 |
| | Residual | 76.75 | 11 | 6.98 | | |
| | Total | 3261.05 | 16 | | | |
| | | $R^2 = 0.9766$, Adj $R^2 = 0.966$, RMSE = 2.632, cv = 2.66% | | | | |
| $CO_2$ selectivity | Model | 198.39 | 5 | 39.68 | 30.16 | 0.0027 |
| | $x_1$ | 76.18 | 1 | 19.04 | 14.47 | 0.0126 |
| | $x_2$ | 109.83 | 1 | 109.83 | 83.49 | 0.0003 |
| | $x_1^2$ | 1.58 | 1 | 1.58 | 1.20 | 0.3231 |
| | $x_1 x_2$ | 2.72 | 1 | 2.72 | 2.07 | 0.2100 |
| | $x_2^2$ | 2.58 | 1 | 2.58 | 1.96 | 0.2203 |
| | Residual | 14.47 | 11 | 1.32 | | |
| | Total | 212.86 | 16 | | | |
| | | $R^2 = 0.9466$, Adj $R^2 = 0.922$, RMSE = 1.016, cv = 3.03% | | | | |

In general, the fitted model could reflect the changes in chlorobenzene degradation efficiency and $CO_2$ selectivity under different flow rates and discharge power. The effect of flow rate and discharge power on the removal efficiency of chlorobenzene is plotted in Figure 9. The effect of chosen process parameters on the performance of the reactor is in agreement with the analysis of Sections 3.2 and 3.3: the removal efficiency of chlorobenzene will increase with discharge power since more active substances can be produced. Moreover, the improved effect of discharge power is more significant for large flow rates since the removal efficiency gradient is not uniform, which indicates that the interactions between discharge power and flow rate are not negligible. Moreover, the ultimate low *p*-value of the term $x_1 x_2$ (<0.01) also supports this conclusion.

The effect of flow rate and discharge power on the $CO_2$ selectivity of chlorobenzene is plotted in Figure 10. The chosen process parameter exhibits a similar effect on the process performance as discussed in Sections 3.2 and 3.3. Moreover, the interactions between the two terms on the reaction performance are insignificant as the $CO_2$ selectivity gradient is almost uniform. The *p*-value of $x_1 x_2$ (>0.05) also supports this conclusion.

### 3.5. Byproducts Analysis and Reaction Mechanism

In this study, the byproducts of chlorobenzene degradation were analyzed by GC-MS and are summarized in Table 4. The aromatic organic compounds include $C_6H_4Cl_2$ and $C_6H_4ClNO_2$. The aliphatic organic compounds mainly contain carboxyl, ether, and halogenated groups, and the length of the carbon chain mainly concentrates between $C_2$ and $C_4$.

**Table 4.** Byproducts analysis results.

| Name | Structure | DBD Reactor | DBD-Catalysis Reactor |
|---|---|---|---|
| Oxalic acid |  | √ * | √ |
| Maleic acid |  | √ | √ |
| Benzene, 1,2 (or 1,3 or 1,4)-dichloro- |  | √ | √ |
| Acetic Acid, Dichloro- |  | √ | √ |
| 2,5-Cyclohexadiene-1,4-dione, 2-chloro- |  | √ | |
| Benzene, 1-chloro-3 (or 2 or 4)-nitro- |  | √ | √ |
| 4-chloro-2-nitrophenol |  | √ | √ |
| Phenol, 2,4-dichloro-6-nitro- |  | | √ |
| Phenol,2,4-dichloro- |  | √ | √ |
| Pyrocatechol, 3,4,6-trichloro- |  | | √ |
| Benzene, 1,1′-oxybis [4-chloro- |  | √ | √ |

*: The detected byproducts were marked by "√".

In previous studies, Zhu et al. thought that the chlorinated aromatic organic compounds would undergo dechlorination first, and then the benzene ring would be opened under the attack of active particles [59]. When analyzing the degradation process of chlorobenzene, Shahna, Haibao Huang considered that OH· radical played an important role in the degradation process of chlorobenzene [60,61]. Liu et al. observed a high proportion of oxalic acid ($C_2$) and maleic acid ($C_4$) when analyzing the degradation products of chlorobenzene in the DBD reactor and attributed the production of oxalic acid ($C_2$) and maleic acid ($C_4$) to the cracking of chlorobenzene under high-energy electron collision. The degradation process of chlorobenzene is divided into three steps: firstly, chlorobenzene breaks down into H atom and chlorophenyl under the attack of high-energy electrons, then chlorophenyl is decomposed into oxalic acid ($C_2$), chloromaleic acid ($C_4$), and other short-chain aliphatic acids under the attack of OH, O, and H. Finally, these short-chain aliphatic acids are further oxidized to $CO_2$ and $H_2O$ [62].

However, the cracking of the benzene ring requires relatively high electron energy. As a result, during the degradation process, a high proportion of the aromatic compounds will keep the benzene ring unopened. Thus, some free radicals in the reaction region will combine with the benzene ring to form various kinds of substitution products. These reactions not only occur in the weak discharge region but also on the surface of the catalyst. Moreover, the probability for the formation of polysubstituted aromatic compounds in the discharge region is relatively low due to the short lifetime of active substances such as OH radicals and Cl atoms. On the surface of the catalyst, the survival time of these active particles is longer than that in the discharge space. Therefore, it is easier to produce aromatic compounds with multiple substituents on the catalyst surface. However, since the toxicity of the aromatic compounds with multiple substituents may be higher than that of the original pollutants, it is vital to avoid the production of such substances and the effect of plasma catalysis on the formation of those harmful byproducts is not negligible.

In this study, 4-chloro-2-nitrophenol, 2,4-dichloro-5-nitrophenol, 2,4-dichlorophenol, biphenyl, and other substances were detected in the byproducts. It was worth noting that the above multiple substituted aromatic compounds could be formed by the same disubstituted aromatic compound 4-chlorophenol with different active substances (OH, Cl and $NO_2$). However, 4-chlorophenol was not detected in the byproducts, while dichlorobenzene and chloronitrobenzene, the reaction product of chlorobenzene with Cl or $NO_2$ were detected. Considering the oxidation potential (2.8 eV) of hydroxyl radical is higher than that of chlorine atom (1.36 eV) and nitro radical (0.96 eV), the possibility of forming the two latter substances alone without forming 4-chlorophenol in the discharge region is relatively low. In addition, the existence of 2-chlorophenyl ether also supports the production of 4-chlorophenol during the reaction process since 2-chlordiphenyl ether is most likely to be produced by the dehydration of two 4-chlorophenols. Furthermore, it is easier for chlorobenzene to produce paradisubstituted compounds during the reaction. On the other hand, the detected polysubstituted aromatic compounds all have two or more inert groups (-Cl, -$NO_2$), which indicates that most of the easily oxidized organic compounds produced in the degradation process are rapidly converted into higher oxidized organic compounds (finally $CO_2$ and $H_2O$). Moreover, these reactions can be considered as the non-rate-determining step for the degradation of chlorobenzene. In this process, the aromatic compounds with multiple substituents formed by the combination of a chlorine atom and nitro radical are not easy to participate in the reaction due to their strong inertia.

In this study, it is speculated that chlorobenzene is firstly attacked by high-energy electrons to form chlorophenyl in the reaction region. Then the chlorophenyl will react with nitro radicals, Cl atoms, and OH radicals to form 4-chlorophenol, dichlorobenzene, and chloronitrobenzene. Dichlorobenzene and chloronitrobenzene will be accumulated in the reaction region due to their low reactivity. The formed 4-chlorophenol is relatively easy to be oxidized, and some of 4-chlorophenol are dechlorinated under the attack of oxidative free radicals, and a ring-opening reaction takes place to form short-chain aliphatic acids such as oxalic acid ($C_2$) and maleic acid ($C_4$). Some of 4-chlorophenol react with Cl atom or

nitro radical to form polysubstituted aromatic compounds such as 4-chloro-2-nitrophenol, 2,4-dichloro-5-nitrophenol, and 2,4-dichlorophenol. Diphenyl chloride is dehydrated from two 4-chlorophenols. Aromatic compounds with multiple substituents (pyrocatechol, 3,4,6-trichloro-, phenol, 2,4-dichloro-6-nitro-) were more easily produced on the surface of the catalyst because the active material adsorbed on the surface of the catalyst has a longer survival time than that in the discharge space. The speculative decomposition mechanism of chlorobenzene is shown in Figure 11.

**Figure 11.** A speculative decomposition mechanism of chlorobenzene by DBD. *: The R · refers to the free radicals generated by the DBD discharge.

## 4. Conclusions

Compared with the single DBD reactor and the DBD reactor coupled with $Al_2O_3$, the DBD reactor coupled with $CuO/\gamma$-$Al_2O_3$ catalyst demonstrated better chlorobenzene degradation and complete oxidation ability. The DBD reactor coupled with $CuO/\gamma$-$Al_2O_3$ catalyst can effectively degrade chlorobenzene into $CO_2$ through the oxidation of intermediates, especially under low voltage conditions. With the increase in energy density, the intermediates of chlorobenzene degradation start to accumulate, and the promotion effect of catalyst is gradually weakened. In addition, the impacts of discharge power and flow rate on the degradation of chlorobenzene were investigated using the RSM method. The

regression correction ($R^2$) coefficients obtained indicates that the regression model fits the experimental results well. The analysis of variance also supports the conclusion that the interactions between flow rate and discharge power for chlorobenzene removal are not trivial. Moreover, based on the analysis of byproducts, chlorobenzene may combine with OH, O, and other free radicals to form 4-chlorophenol under the attack of high-energy electrons. The latter will be broken into unsaturated hydrocarbons with four or two carbon atoms after further oxidation and ring-opening. These unsaturated hydrocarbons can be fully oxidized under the attack of free radicals such as OH· and O. Although the combination of DBD with CuO/γ-Al$_2$O$_3$ catalyst has many advantages, it may lead to the formation of extra polychlorinated compounds.

**Author Contributions:** Conceptualization, X.J., G.W.; Data curation, X.J.; Formal analysis, X.J. and R.Z. (Ruina Zhang); Funding acquisition, G.W. and S.C.; Investigation, X.J.; Methology, R.Z. (Renxi Zhang) and G.W.; Project administration, R.Z. (Renxi Zhang); Resources, S.C. and J.H.; Supervision, R.Z. (Renxi Zhang) and W.Z.; Validation, L.L. and W.Z.; Visualization, X.J. and F.G.; Writing—orginal draft, X.J.; Writing—review and editing, X.J. and R.Z. (Renxi Zhang). All authors have read and agreed to the published version of the manuscript.

**Funding:** This research was funded by the National Natural Science Foundation of China (no. 21577023), the technology innovation and level promotion project supported by Shanghai State-owned assets supervision and Administration Commission (no. 2018001) and the science and technology innovation action project supported by the Science and Technology Commission of Shanghai Municipality (no. 18DZ1202605) and Natural Science Foundations of Hainan Province (no. 219QN290).

**Institutional Review Board Statement:** Not applicable.

**Informed Consent Statement:** Not applicable.

**Data Availability Statement:** The data presented in this study are available on request from the corresponding author. The data are not publicly available due to at this time as the data also forms part of an ongoing study.

**Acknowledgments:** The authors thank Liuyanghai Chao for his help in the research work.

**Conflicts of Interest:** The authors declare no conflict of interest.

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
