# Peer review of "Chlorobenzene Removal Using DBD Coupled with CuO/γ-Al2O3 Catalyst"

_applsci, doi:10.3390/app11146433_

Round 1

Reviewer 1 Report

This paper describes metal catalytic decomposition of chlorobenzene. The authors demonstrated catalytic activity of DBD with CuO/γ-Al2O3.  And the fate of chlorobenzene after decomposition is well-cheracterized (including ozone production). The findings in this paper are of basic interest and tinformative for several fields of chemistry, I think. Therefore, I recommend publication in Applied Science after correction of the following points.

  • Title: Copuled -> Coupled

Reviewer 2 Report

This original manuscript presents interesting results concerning the process of chlorobenzene degradation in the dielectric barrier discharge reactor coupled with supported CuO/gamma-Al2O3 catalyst. The paper is well written and sufficiently illustrated. The scope of research is suitable for publication in the Applied Sciences. However, the minor changes should be done to prepare the manuscript for publication (Minor Revision needed).

Questions:

1) Page 8, Lines 266-285: How the concentration of ozone was measured? Please, specify in the Experimental Section.

2) Page 12, Table 4: Can authors tell the readers what are the main byproducts among the listed in Table 4, and what their total concentration (or selectivity, %)? Does the catalytic path have any advantage in this aspect?

Comments:

1) Page 2, Line 87: “.. was selected as substitute” should be probably replaced by “… substrate”.

2) Page 3, Line 106: “Bruner-Emmet-Teller” should be replaced by “Brunauer-Emmet-Teller”.

3) Page 5, Table 1: The title for Table 1 is obviously wrong and must be changed.

4) Page 5, Line 189: The name of Section 3 must be “Results and Discussion” instead of “Experimental Section”.

5) Page 6, Line 198: The units (degrees) should be added to the title of the x-axis (2 Theta).

6) Page 6, Line 212: Eliminate the repeated word (that that).

7) Page 7, Figure 5: The y-axis shows the CO selectivity (%), but not the Removal Efficiency of Chlorobenzene, as it is stated in the figure caption.

8) Page 8, Line 281: “can improves” should be replaced by “can improve”.

9) Page 9, Lines 312 and 315: “chemical bend(s)” should be replaced by “chemical bond(s)”.
